# A Malware Distribution Simulator for the Verification of Network Threat Prevention Tools

**DOI:** 10.3390/s21216983

**Published:** 2021-10-21

**Authors:** Song-Yi Hwang, Jeong-Nyeo Kim

**Affiliations:** 1Department of Information Security Engineering, University of Science and Technology (UST), Daejeon 34113, Korea; syhwang@ust.ac.kr; 2Electronics and Telecommunications Research Institute, Daejeon 34129, Korea

**Keywords:** IoT malware, propagation, diffusion, tool, verification

## Abstract

With the expansion of the Internet of Things (IoT), security incidents about exploiting vulnerabilities in IoT devices have become prominent. However, due to the characteristics of IoT devices such as low power and low performance, it is difficult to apply existing security solutions to IoT devices. As a result, IoT devices have easily become targets for cyber attackers, and malware attacks on IoT devices are increasing every year. The most representative is the Mirai malware that caused distributed denial of service (DDoS) attacks by creating a massive IoT botnet. Moreover, Mirai malware has been released on the Internet, resulting in increasing variants and new malicious codes. One of the ways to mitigate distributed denial of service attacks is to render the creation of massive IoT botnets difficult by preventing the spread of malicious code. For IoT infrastructure security, security solutions are being studied to analyze network packets going in and out of IoT infrastructure to detect threats, and to prevent the spread of threats within IoT infrastructure by dynamically controlling network access to maliciously used IoT devices, network equipment, and IoT services. However, there is a great risk to apply unverified security solutions to real-world environments. In this paper, we propose a malware simulation tool that scans vulnerable IoT devices assigned a private IP address, and spreads malicious code within IoT infrastructure by injecting malicious code download command into vulnerable devices. The malware simulation tool proposed in this paper can be used to verify the functionality of network threat detection and prevention solutions.

## 1. Introduction

The Internet of Things (IoT) has been incorporated into many industries, including home appliances, healthcare, manufacturing, and production [1]. As a result, the demand for IoT devices has been increasing in the industry [2,3]. In 2020, 11.7 billion IoT devices were connected to the Internet. These account for 54% of the 21.7 billion active connections worldwide [4]. More than 30 billion IoT devices are expected to be connected to the Internet by 2025 [5,6]. As demand for IoT devices has increased, different hardware and OSs, with diverse communication protocols, standards, and software applications have been created [7]. This large, nonuniform IoT framework has introduced a variety of new security challenges. In other words, with the increase in the number of IoT devices, IoT infrastructure security incidents have also increased [8]. According to this trend, if the number of IoT devices connected to the network increases while the security of IoT devices is still low, the number of cyberattacks targeting IoT devices will also increase, and the damage will be significant.

Each IoT device with a low-security level has become a potential security threat to the IoT infrastructure, resulting in security incidents such as IoT malware attacks, denial of service attacks, and the leakage of personal and confidential information. The reason IoT devices continue to be targeted by attackers is that the security level of IoT devices is still very low. Current IoT devices are not only vulnerable to older attack techniques but also easily exposed to new attacks such as IoT malware attacks targeting specific IoT devices. Several IoT infrastructure protection technologies are currently being studied to prevent and defend against possible security incidents in the IoT environment. As security threats become more diverse, a variety of security solutions, including intrusion detection, cryptography algorithms, DDoS attack detection, and hardware-based isolation, have been proposed to reduce the damage caused by attacks [9]. However, even if an effective security solution has been developed, it should not be applied to the real world without verifying the functionality and performance of the security solution. This is because there is no certainty that a security solution can prevent threats. Therefore, the reliability must be demonstrated by testing the functionality of the security solution using attack tools that can cause the same threats as real attacks.

In particular, as the source code of Mirai malware was released on the Internet, cyber attackers have created and distributed numerous Mirai variant malware to infect IoT devices [10]. In 2019, nearly 440,000 malware variants were found, 1200 per day [11]. In addition, IoT malware attacks are up 5% from 2019, with 34.3 million cases recorded in 2020 [12]. This figure shows that Mirai-modified malware is used a lot to threaten IoT devices. Therefore, using Mirai botnet for research purposes is suitable for verifying security solutions that detect or block malware within IoT infrastructure. However, Mirai targets IoT devices around the world to infect as many devices as possible and cause massive DDoS attacks. Mirai’s feature of spreading malware to global environments makes it difficult to verify security solutions that protect certain infrastructure or local environments. Because Mirai botnet cannot spread malware in a local environment, it cannot verify security solutions that detect or prevent malware from spreading within a specific infrastructure. Therefore, in this paper, we implement a modified Mirai botnet, called a malware simulation tool, to allow malware to spread even in local environments, and a detailed description is covered in Section 3.

In this paper, we propose an attack tool to verify the security solutions that detect and protect spreading malware within IoT infrastructure. The main contributions of this paper are to:Analyze and evaluate the propagation techniques used by the Mirai malware.Explain why the Mirai malware is not suitable to verify the functionality of the security solutions and propose a method to propagate malware to IoT devices that are assigned private IP addresses.Evaluate new propagation technology in small network environments based on total propagation time.Define the real and virtual experimental environments that can verify the functionality and performance of network-based threat prevention solutions.Propose the optimal conditions that are to be set for a malware simulation tool, such as a scanning method, the number of master bots, and the number of scan packets, for rapid malware spreading. The value to set for the malware simulation tool can be determined by comparing the number of IoT devices used in the experiment with the number of IoT devices in the infrastructure to which the security solution will be applied.

The remainder of this paper is organized as follows. Section 2 analyzes and compares several IoT malware, and explains why the malware simulation tool was implemented based on Mirai. In Section 3, we discuss in detail how the Mirai malware, which is the foundation of the malware simulation tool, spreads malware. This section provides a reference to comparing the differences between the Mirai malware and the malware simulation tool with respect to a scanning strategy to search for vulnerable devices and a propagation strategy to inject malicious code into vulnerable devices. In Section 4, we explain why Mirai cannot be used to verify security solutions that detect and block threats spreading within IoT infrastructures. In addition, unlike the Mirai malware, we describe two methods a malware simulation tool can propagate a malicious code to vulnerable IoT devices even that they have been assigned private IP addresses. Section 5 compares the time it took for the malware simulation tool to infect every device used in the experiment with malicious code, depending on the scanning methods, the number of bots performing scans, and the number of packets transmitted per port scan. Finally, in Section 6, we conclude with the implications of the research and future works.

## 2. Related Works

### 2.1. Malware Detection Techniques

Lim et al. [13] proposed a smart segmentation framework with attention to the usage of the Domain Generation Algorithm (DGA) in the botnet malware. The smart segmentation framework detects the botnet malware by considering the behavior of querying domain names other than the reference, which is domain names with those of servers normally operated at IoT services, ones as botnet malware’s behavior. Because the C&C server continues to change its IP address to hide its location or avoid detection and prevention, malware does not explicitly specify the IP address of the C&C server. On the other hand, most IoT devices mainly act as sensors or actuators, IoT devices explicitly specify the IP address of the target that received control commands or transmitted the collected data. Therefore, the smart segmentation framework suspected that a connection request outside of target via a DNS query request is the behavior of malware attempting to connect to the C&C server. Moreover, they used the Mirai malware to test the functionality of the smart segmentation framework. They configured a testbed to test whether the smart segmentation framework could block or mitigate the spread of threat, and operated Mirai malware to spread threats in the testbed.

Kumar et al. [14] proposed a algorithm to detect bots infected with Mirai or Mirai-like IoT malware. They used Mirai traffic signatures and a two-dimensional sub-sampling approach for the bot detection algorithm, and studied the relationship between average detection delays and sampling frequencies to identify vulnerable and non-vulnerable devices. They implemented a testbed that simulates the actual behavior of Mirai malware to collect Mirai traffic signatures used to create the bot detection algorithm. They simulated the actual behavior of Mirai malware on the testbed and collected packets created by Mirai malware and analyzed the unique signatures of the Mirai malware to identify the existence of Mirai malware on IoT devices. The analyzed information obtained by simulating Mirai’s actual behavior was used to design the bot detection algorithm that proposed by them to detect IoT malware bots similar to Mirai.

Liu et al. [15] proposed an integrated architecture for IoT malware analysis and detection. They do not propose a technical method to detect and block IoT malwares. However, they suggest IoT malware analysis methods. The integrated architecture they proposed includes static analysis, dynamic analysis, evolution analysis, and network traffic analysis. They performed static and dynamic analysis with Mirai malware to demonstrate the integrated architecture. Furthermore, they performed an evolution analysis to analyze the similarities and differences between the Mirai variants and Mirai. In addition, by collecting traffic data of two IoT malware, they analyzed the destination port number used most often and the regional distribution of infections.

Meidan et al. [16] proposed a network-based anomaly detection method for the IoT. Their method immediately isolates the compromised IoT device from the network if attack related anomalies are detected. They used deep autoencoders to detect anomalies. They used Mirai and BASHLITE to evaluate their method. They infected IoT devices by running Mirai and BASHLITE binaries. They collected network traffic before and after malware infections generated by IoT devices and used it as training data of deep autoencoders.

### 2.2. IoT Malwares

We analyze the features of IoT malwares such as BASHLITE, Hajime, Mirai, BrickerBot, Reaper, Persirai, Satori, VPNFilter, and EchoBot [17,18,19]. Table 1 and Table 2 compare each feature of IoT malwares.

BASHLITE—2014. Similar to Mirai, BASHLITE’s goal is to conduct DDoS attacks by infecting ARM and MIPS architecture-based Linux IoT devices. Part of the BASHLITE’s source code had been released, resulting in a large number of BASHLITE-modified malware, including Mirai. BASHLITE’s scanning method for finding vulnerable IoT devices is a random scanning method. Furthermore, BASHLITE’s malware infection method is either one CVE (Common vulnerabilities and exposures) vulnerability attack or a dictionary attack to infect vulnerable IoT devices with malware. The botnet architecture of BASHLITE is a centralized structure that manages botnet on C&C servers. The C&C server and botnet use a dedicated protocol to communicate with each other. BASHLITE has a characteristic that is closing the port used as the target port for infection to prevent other malware from re-infecting vulnerable devices infected by BASHLITE.

Hajime—2016. Hajime attacks all devices based on the ARM, MIPS, and x86/64 architecture connected to the Internet. However, Hajime was mainly detected digital video recorders (DVRs), web cameras, and routers. Hajime does not appear to present any malicious behavior. The scanning method of Hajime is a random scanning method. Moreover, the malware infection method of Hajime is a dictionary attack. Unlike BASHLITE and Mirai, The architecture of Hajime is a peer-to-peer (P2P) structure and does not have a C&C server. Hajime has characteristics including ports closing, binary removal, anti reboot, victim architecture’s scan, process masquerade, code’s modularity or update system, and another botnet removal.

Mirai—2016. Mirai attacks MIPS, ARM, and x86/64 architecture-based Linux IoT devices, and aims to perform DDoS attacks. Mirai has different types of flood attacks, such as Syn Flood, UDP Flood, ACK Flood, Push flood, HTTP Flood, DNS Waterboarding, and DNS Amplification. The scanning method of Mirai is a stateless random scanning method. The stateless random scanning method is that does not wait for a timeout before moving on to another random IP address to scan. Therefore, the stateless scan is far more rapid rather than the ordinary scanning method. Also, the infection method of Mirai is a balanced dictionary attack. The balanced dictionary attack is that botnet selects a random subset of credentials from the dictionary and only attempts those credentials. The balanced dictionary attack determines the subset of credentials based on the most frequently used default for the device. The botnet architecture of Mirai is a centralized architecture that uses a dedicated communication protocol. Mirai has several features: process masquerade, binary removal, ports closing, other botnet removals, anti reboot, and victim architecture’s scan. Most notably, the source code of Mirai has been released on the Internet.

BrickerBot—2017. BrickerBot attacks embedded Linux-based IoT devices that include Busybox software. However, BrickerBot does not use the bot for DDoS attacks. BrickerBot finds vulnerable IoT devices and makes them destroy. Unlike other malware that aims to launch DDoS attacks by creating the botnet as large as possible, BrickerBot does not self-replicate in infected bots. It is the difference between other malware and BrickerBot. BrickerBot uses the stateless random scanning method to scan the vulnerable IoT device, and the balanced dictionary attack to infect the vulnerable device with malware. Unlike BASHLITE and Hajime, BrickerBot performs the balanced dictionary attack, which randomly selects the weight in the dictionary and uses credentials with weights above the selected weight. BrickerBot’s botnet has a centralized architecture same as Mirai’s structure.

IoT Reaper—2017. IoT Reaper attacks IoT devices from certain manufacturers. The devices infected with IoT Reaper were used in DDoS attacks. IoT Reaper’s scanning method is a random scanning method. Moreover, IoT Reaper’s malware infection method is various attacks using CVE such as remote code execution vulnerability attack and web authentication bypass vulnerability attack. The architecture of IoT Reaper is also the centralized structure just like BASHLITE, BrickerBot, and Mirai. IoT Reaper has seven features: process masquerade, binary removal, ports closing, other botnet removals, anti reboot, code’s modularity or update system, and victim architecture’s scan.

VPNFilter—2018. VPNFilter attacks specific manufacturers’ device-based ARM, MIPS, x86/64 architectures. VPNFilter causes various attacks such as PDoS, firewall DoS, data leakage, and intermediate attacks. VPNFilter uses the stateless scanning method for a more rapid scan. VPNFilter has attack vectors using various CVE vulnerabilities to infect vulnerable devices with malware. Similar to the architecture of BASHLITE, BrickerBot, IoT Reaper, and Mirai, the VPNFilter is the centralized structure. Uniquely, the VPN filter changes the firmware of the vulnerable device, so the vulnerable device remains infected even if reboot. Also, the VPNFilter has characteristics such as code’s modularity or update system, anti reboot, and victim architecture’s scan. Furthermore, VPNFilter uses the Domain Generation Algorithm (DGA) that randomly generates domain addresses for the purpose of avoiding C&C server detection.

As the source code of Mirai was released on the Internet, Mirai variants have been continuously reported [21]. Therefore, the malware simulation tool was implemented based on the Mirai that is highly referenced by attackers.

## 3. The Foundation of the Malware Simulation Tool: Mirai Source Code Released on the Internet

### 3.1. Propagation Technique of the Mirai Malware

As shown in Figure 1, the Mirai botnet infrastructure consists of infected devices called Mirai bots, a C&C (Command and control) server that commands and controls Mirai bots, a report server that collects information about the IoT device to infect, a DNS server that supplies the public IP address of the C&C server and the report server, and a malware loader that injects the Mirai malware into IoT devices.

The Mirai botnet performs two main actions: The first is to find and infect vulnerable IoT devices. The second is to launch a DDoS attack on the target. The more vulnerable IoT devices infected, the larger the size of the Mirai botnet and the greater the DDoS attack. Consequently, Mirai malware seeks out as many vulnerable IoT devices as possible around the world. Therefore, the propagation technique applied by Mirai is the stateless random scanning method and the malware loader’s malware infection method, and a detailed description of the malware loader’s malware infection method is covered in Section 3.1.2.

#### 3.1.1. Stateless Random Scanning Method

Mirai performs TCP SYN half open scanning method. Mirai sends the SYN packet to the target to check the status of the target port and waits for a response. When a SYN/ACK response is received from the target, the target port is opened, and when a RST/ACK is received, the target port is closed. Depending on what response the target sent, the port status can be determined. However, if a SYN/ACK response is received from the target, Mirai immediately tears down the connection by sending a RESET. It is called stateless scanning or stealth scanning because it does not remain connection.

Mirai malware seeks to infect as many IoT devices as possible. This is because the larger the Mirai Botnet, the more powerful the DDoS attack can be. Thus, Mirai malware applies a stateless random scanning method to scan all IP addresses, except for specific IP addresses in Table 3, which cannot be assigned to a device. However, the blacklist includes internal networks. A device can be assigned a private IP address, but the problem is that the malware loader cannot connect to the device located on the internal network. Therefore, Mirai does not scan an internal network.

#### 3.1.2. The Malware Loader’s Malware Infection Method

In Mirai malware, the roles of the Mirai bot and malware loader are precisely distinguished. The Mirai bot scans vulnerable devices, and the malware loader injects malware into vulnerable devices. Mirai malware goes through six stages of the propagation process to infect IoT devices, as shown in Figure 2. The Mirai bot performs steps 1 to 3, and the malware loader performs steps 4 to 6.

Step 1. Scan telnet port. The Mirai bot scans devices that use telnet services. The bot sends packets with the destination port number set to 23 or 2323, and the destination IP address is randomly set. Then, it checks the response to determine whether the telnet service is enabled on the device. Mirai malware uses TCP half open scanning to avoid establishing TCP connections with the victim device. Therefore, there are no logs left on the victim’s device, which allows stealth scanning.

Step 2. Attempt credential brute force attack on the telnet login. Mirai Bot attempts to connect to victim devices that enable the telnet service. It uses 62 credentials stored in Mirai malware to launch brute force attacks for up to 30 s for a telnet login.

Step 3. Report information of victim devices. When the telnet login is successful, the Mirai bot reports the IP address, port number, and telnet service login account of a victim device to the report server called scanListen. Therefore, the report server leaves the 48,101 port open and receives information on the victim device passed on by the Mirai bot.

Step 4. The report server delivers information to the malware loader. The report server passes the information on the victim device received from the Mirai bot to the malware loader. That information is passed to the malware loader in the order it is passed from the Mirai bot to the report server. There are only two tasks that the report server performs. The report server receives information from the Mirai bot and sends it to the malware loader.

Step 5. Check the status of the victim device. Pre-investigation is conducted to inject Mirai malware into victim devices. The malware loader connects to the victim device using the information received from the report server. It then collects the victim devices’ information, such as CPU architecture, type of shell script, and checks whether Busybox and Wget/TFTP are installed, which is necessary to inject malicious code.

Step 6. Inject Mirai malware. The malware loader injects Mirai malware into the victim’s device. Based on the information gathered from the victim’s device, the malware loader selects the appropriate program to inject Mirai malware and the suitable Mirai malware binary for the CPU architecture.

#### 3.1.3. Domain Generation Algorithm (DGA)

Most botnet malware hardcoded the IP address of the C&C server. This is because the C&C server has to communicate with the bot periodically to control and command the bot. However, the hardcoded IP address of the C&C server was easily detected and blocked by the security manager. A botnet malware evolves to hide the location of C&C servers and to avoid the defense solution to detect and prevent the hardcoded IP addresses or URLs. Each bot uses the domain generation algorithm (DGA) to generate a predefined number of random domain name each day, and attempt to connect to the C&C server with each domain name.

Initial Mirai did not use DGA. However, there have been numerous variants since the Mirai source code was released on the Internet [22]. There is also variant of the Mirai botnet family, including DGA [23]. The Mirai source code released on the Internet does not include DGA. However, as shown in Figure 3, Mirai hardcodes the domain name instead of the IP address of the C&C server.

As a result, although Mirai bot does not attempt to connect with the C&C server with many domain names created by DGA, the Mirai Bot must request a domain query to the DNS server to communicate with the C&C server.

## 4. Malware Simulation Tool

The malware simulation tool proposed in this paper is a simulation tool to rapidly propagate malware in a particular infrastructure or local network environment. This tool verifies the functionality and performance of network-based malware detection and threat prevention solutions that detects and blocks spreading threats within the IoT infrastructure. It is especially suitable for use in verifying the smart segmentation framework [13] among techniques to prevent malware propagation attacks. We consider that the malware attack is a kind of threats spreading in the network Therefore, the malware simulation tool is implemented based on Mirai malware, which informed the seriousness of IoT security and also was released its source code on the Internet. However, the propagation technique of the Mirai malware cannot spread malware into internal networks.

### 4.1. Problems with the Propagation Technique of the Mirai

In order to verify security solutions that detect and block malware spreading to IoT infrastructure, malware has to be spread in IoT infrastructure. However, the propagation technique of Mirai does not scan internal networks, so Mirai cannot spread malware in IoT infrastructures managed by security solutions. When using Mirai Malware’s propagation techniques, there were five problems.

#### 4.1.1. Wide Range of IP Addresses to Scan

The random scanning method used by Mirai malware is not suitable for targeting specific infrastructure with a limited range of IP addresses. The random scanning method randomly selects one of the Internet IPv4 addresses in the world and scans the devices to which the corresponding IP address is assigned. However, more than 3.6 billion IPv4 addresses are currently assigned worldwide [24,25]. Assuming that a malware propagation attack is performed on a particular infrastructure, using a random scanning method that scans 3.6 billion IP addresses will take a long time to infect the entire device located on a particular infrastructure. Eventually, Mirai malware cannot propagate quickly within the infrastructure using a random scanning method.

#### 4.1.2. Inaccessible Internal Network

Mirai malware does not scan internal networks. The malware loader must establish network communication to inject malware binaries into the victim device. However, as shown in Figure 4, the malware loader cannot access the devices to which the private IP addresses are assigned. Thus, excluding loopback, internal networks, and multicast addresses, Mirai malware attempts to attack only devices that have been assigned public IP addresses.

#### 4.1.3. Duplicate Infections

When the number of scanning packets is high, multiple bots simultaneously perform the propagation process by targeting one device. Because Mirai malware has no limit to the range of IP addresses to scan and randomly selects IP addresses, it is unlikely that multiple bots will infect the same device. However, if 160 scanning packets are exported per bot within the internal network with a limited range of available IP addresses, multiple bots are likely to infect the same device.

There is an interval between the time a vulnerable device downloads malware and the time the downloaded malware kills the telnet service on the vulnerable device. Even if the vulnerable device has downloaded the malware, other bots can scan this vulnerable device during the interval because the telnet service is still activated. As a result, the victim’s device is still in a status that can be infected until the first-downloaded malware kills the telnet service on that device. This means that the vulnerable device is likely to download malware several times. The problem is that Mirai malware has a killer process that detects and terminates malware other than itself. Therefore, all malware downloaded during the time interval on the victim’s device runs in turns and then terminates. As shown in Figure 5, if two malware terminate each other at the same time after the Telnet service is disabled, the current attack method using the Telnet service vulnerability cannot re-infect the device. This means that the more duplicate infections occur, the fewer devices the attacker can control.

#### 4.1.4. Network Infrastructure Overload

When all bots in the botnet perform scanning actions, network congestion traffic increases and network equipment is overloaded. This is because all packets sent for port scanning during malware attacks in the internal network go through the internal network’s equipment. If the number of bots is small, the network equipment processes the packets exported by the bots without delay. However, as the number of bots increases, the number of packets that network equipment must process increases. Eventually, network equipment cannot operate normally if the traffic exceeds what the network equipment can handle.

While scanning a telnet port, the Mirai malware transmits 160 packets. In fact, Mirai generates 160 packets for port scanning until the Mirai process is terminated. If the malware simulation tool, as in Mirai malware, sets the number of exporting network packets per telnet port scan to 160, the gateway must handle network packets that increase by a factor of 160 as the number of bots increases. Malware attacks on IoT devices in a specific infrastructure or internal network can become a DDoS attack on network devices in the infrastructure. In this case, owing to a decrease in the performance of the gateway, the propagation speed of the malware slows down, or the gateway fails to perform normally. This prevents malware from spreading, and the attacks are more likely to be detected by security solutions.

#### 4.1.5. Inefficient Infrastructure Deployment

Botnet infrastructures, such as C&C servers, report servers, and malware loader, must be configured in an internal network that uses private IP addresses to propagate malware. However, this approach is inefficient, requiring a botnet infrastructure to be built for each infrastructure under attack. Another problem is that the IP address of the network layer header is not referenced in the process of malware propagation. Internal network communication refers only to the MAC address of the data-link layer header. Consequently, this causes the problem in which a security solution that only detects security threats by analyzing network layer traffic cannot detect malware attacks. The malware simulation tool proposed to address these problems applies a limited range scanning method and a D2D command injection propagation method. These methods enable high-speed malware propagation even in IoT devices that are assigned private IP addresses within internal networks. Both the limited range scanning method and the D2D command injection propagation method are described in detail below.

### 4.2. Solutions to the Problems of the Mirai Malware’s Propagation Technique

To address the problem that Mirai is difficult to spread malware within a particular infrastructure, we propose the malware simulation tool with a device-to-device command injection method and with restricted-range stateless scanning methods. As a result of applying the device-to-device command injection method to the malware simulation tool, it can be propagated malware to IoT devices that have been assigned private IP addresses within internal networks (Section 4.1.2), solve inefficiency in Botnet infrastructure construction (Section 4.1.5). Furthermore, as a result of applying restricted-range stateless scanning methods to the malware simulation tool, it can quickly spread malware within specific target infrastructures by resolving the problem of wide range of IP addresses to be scanned (Section 4.1.1). Therefore, the malware simulation tool makes it possible to verify the functionality of security solutions by spreading malware to IoT devices assigned private IP address.

#### 4.2.1. Device-to-Device Command Injection Method

The malware simulation tool uses the device-to-device (D2D) command injection (CI) propagation method. In the D2D CI propagation method, the bot injects file transfer commands into vulnerable IoT devices. Then, these devices execute injected commands to download the malware binary from the server.

This is because, as shown in Figure 4, the malware loader cannot access devices with assigned private IP addresses. For the malware loader to access the devices on the internal network, port forwarding must be set up for each internal network device, or the malware loader must be on the same network as the victim device. Otherwise, the malware loader cannot access internal network devices that use private IP addresses. This is why Mirai malware excludes internal networks from the scanning target.

To address the problem of the Mirai malware infection method, we apply the device-to-device command injection method to the malware simulation tool. Compared to the infection method of Mirai malware, the malware simulation tool does not use the report server. In addition, the bot takes over the role of the malware loader in the malware simulation tool. In other words, the device-to-device command injection method, which is applied in the malware simulation tool, allows the bot to inject file download commands into vulnerable devices and execute injected commands by vulnerable devices to request and download malware from the HTTP server. To be more specific, if telnet login is successful on a vulnerable device, Mirai and the malware simulation tool collect the information, such as shell type, architecture, whether the Wget is installed, about the vulnerable device. Mirai passes the information collected from the vulnerable device to the malware loader, but the malware simulation tool injects the malware download command into the vulnerable device. The malware simulation tool supports cross-compile to create malware binaries that can run on ARM, MIPS, and x86/64 architectures. Moreover, these malware binaries are stored on an HTTP server. Therefore, a vulnerable device can download a malware binary that matches its architecture by using the Wget tool from the HTTP server. In other words, unlike Mirai, the malware simulation tool can spread malware in internal networks because internal network devices access HTTP servers located on the parent network. As bots or vulnerable devices communicate with the C&C server and the HTTP server outside the private network, It can be used to verify solutions that prevent network-based threats.

The malware simulation tool infects vulnerable IoT devices through six propagation stages, as shown in Figure 6. The device-to-device command injection method of the malware simulation tool does not use a report server nor the malware loader.

Step 1. Scan telnet port. The bot scans devices that use telnet services. The bot sends packets by setting the destination port numbers to 23 or 2323. Then, it checks the response to determine whether the telnet service is enabled on the device. The telnet port scanning method was the same as that used by the Mirai malware.

Step 2. Attempt credential brute force attack on the telnet login. The bot attempts to connect to the victim devices that enable telnet service. It uses 62 credentials to launch credential brute force attacks for up to 30s for a telnet login.

Step 3. Gathering information about vulnerable devices. If Telnet login is successful, a preliminary investigation is conducted to inject malware into vulnerable IoT devices. The bot collects the information required to inject malware, such as the CPU architecture and shell-type of vulnerable devices.

Step 4. Inject commands to download malware binary. The bot injects commands to allow vulnerable devices to download malware binaries. Based on the information collected, the bot determines the malware binary with the same architecture as the architecture base of the vulnerable device.

Step 5. Request malware downloads. The vulnerable device executes the commands injected by the bot. The vulnerable device uses the Wget command to request malware download to the HTTP server where the malware binary is stored.

Step 6. Inject malware. The malware loader responds to the request from the vulnerable device and injects the malware binary.

#### 4.2.2. Restricted Range Stateless Scanning Method

Botnets employ several scanning methods to scan the networks in order to detect new devices and infect them [17]. A hardcoded hit-list scanning method, a network class automatic hit-list scanning method, and a random scanning method were scanning methods of actual IoT botnets. The hardcoded hit-list scanning method configures a preprogrammed list of IP addresses for botnet malware to scan. Later, attackers have evolved a scanning method by incorporating code that generates a list of potential targets from a given subnet in the botnet malware’s code. In 2012, botnet malware began to use a random scanning method, which is easier to implement. As shown in Table 1 and Table 2, BASHLITE, Hajime, Mirai, BrickerBot, IoT Reaper, and VPNFilter employ the random scanning method.

Based on the network class automatic hit-list scanning method, which generates a list of potential targets on a given subnet, we propose an evolved scanning method that automatically obtains a subnet mask of a bot performing scan behavior and generates a list of IP addresses to scan. The evolved scanning method called a restricted range stateless scanning method scans network regions built with private IP addresses. The restricted range scanning method scans only the range of IP addresses that can be assigned to devices that are connected to the local network. The restricted range scanning method uses the bot’s local IP address and subnet mask to obtain an allocatable range from the local network that contains the bot. For example, if a bot is assigned a local IP address of 192.168.0.100, and the subnet mask is 255.255.255.0, a total of 254 IP addresses can be allocated from 192.168.0.1 to 192.168.0.254 from the local network that contains the bot. In this case, the bot can scan 254 IP addresses between 192.168.0.1 and 192.168.0.254. The restricted range scanning method means that the bot scans only the range of IP addresses that can be allocated from the local network, which contains the bot. Unlike Mirai, which scans all IP addresses except the blacklist (Table 3), the malware simulation tool has a limited range of IP addresses to scan. Therefore, in this paper, the method of scanning only the range of allocatable IP addresses on the local network that contains the bot is called the restricted range scanning method.

The restricted range scanning method shows fast propagation speed compared to Mirai’s random scanning method because there are not many IP addresses to scan. The propagation speed mentioned in this paper refers to the speed at which malware spreads to IoT devices that make up a specific IoT infrastructure. The malware simulation tool implements to verify network traffic-based malware of threat spread detection or prevention solutions, especially the smart segmentation framework. In order to verify security solutions that detect and block rapidly spreading threats within IoT infrastructure, malware must be spread rapidly within IoT infrastructure. To do this, the restricted range stateless scanning method, which is employed in the malware simulation tool, performs a stateless scan and does not wait for a timeout before moving on to a new IP. As a result, the malware simulation tool can show that the botnet size is growing rapidly.

We propose four scanning methods to find out which scanning method, sequential scanning method or random scanning method, can spread malware faster: a sequential scanning method, a divided and sequential scanning method, a restricted random scanning method, and a divided and random scanning method.

Sequential scanning method. The sequential scanning method scans from the beginning to the end of the limited IP address range. When using this method, the master bot’s local network IP address and subnet mask are needed for obtaining the scanning range. The network ID and host ID are distinguished from the master bot’s local network IP address to obtain a range of IP addresses to which the device can be assigned. This range of IP addresses is the scannable IP address range that the master bot scans. For example, if the master bot’s local network IP address is 192.168.1.101/24, the network ID is 192.168.1.0, and the host ID is 101. The available IP addresses of the private network, including the master bot, range from 192.168.1.1 to 192.168.1.254. The master bot scans from the beginning to the end of the available address in sequence. If the IP address scanned by the master bot is beyond the available address range, it returns to the beginning.

Divided and sequential scanning method. The divided and sequential scanning method scans sequentially from the beginning to the end of a limited IP address range, similarly to the sequential scanning method. The difference is that this scanning method divides the limited range of IP addresses into several zones and allows each master bot to scan different zones. Therefore, as many master bots are required to scan each area as the number of zones is created by dividing the range of available addresses. The divided and sequential scanning method uses modular arithmetic to divide the limited range of available IP addresses into multiple zones. The master bot divides the host ID of the IP address by the number of partitioned areas. The master bot then determines the IP address to scan with the remaining values obtained from the calculations above. For example, if the range of available addresses is divided into two zones, two master bots are needed. Both master bots calculate the remaining values sequentially from the first to last IP addresses in the range of available IP addresses. One master bot scans only IP addresses that the remaining value of 1 when the host ID of an IP address is divided by 2. The other master bot scans only the IP addresses that the remaining value of 0 when the host ID of an IP address is divided by 2.

Restricted random scanning method. The restricted random scanning method randomly selects IP addresses from a limited range of IP addresses. As with other scanning methods, a range of scannable addresses is obtained from the master bot’s local network address. Mirai malware also uses a random scanning method, but there is no limit to the range of scannable IP addresses, allowing access to vulnerable IoT devices around the world. However, the restricted random scanning method is only possible for a given range of available IP addresses. The advantage of the restricted random scanning method is its ability to access devices in the same network quickly, although the number of accessible devices is limited.

Divided and random scanning method. As described by the divided and sequential scanning method, this method also divides the restricted range of available IP addresses into multiple zones and causes each master bot to scan different zones. When a restricted range of IP addresses is divided into two zones, one master bot scans only IP addresses that the remaining value of 1 when the host ID of an IP address is divided by 2. The other master bot scans only the IP addresses that the remaining value of 0 when the host ID of an IP address is divided by 2. Similar to the divided and sequential scanning method, the master bot uses a modular operation to distinguish the zones. However, this scanning method randomly selects IP addresses within a restricted range. The master bot determines whether to scan based on the remaining values when the host ID of the randomly selected IP address is divided by the number of partitioned areas.

## 5. Experiments and Evaluations by Changing the Number of Master Bot and Scan Packet

Problems corresponding to Section 4.1.1, Section 4.1.2, and Section 4.1.5 described in Section 4.1 can be solved by the device-to-device command injection method and the restricted range state scanning method described in Section 4.2. As shown in Figure 7 and Figure 8, it can be seen that malicious code spreads to internal networks with the malware simulation tool.

However, there are still two problems left. The Mirai’s propagation technique increases the number of duplicate infections due to the number of scan packets that do not take into account the scope of the IP addresses to be scanned in Section 4.1.3 In addition, excessive scanning operation packets due to scanning of all bots on the same local network cause network equipment overload problems in Section 4.1.4 To solve these two problems, we need to find the appropriate number of scan packets and the number of master bots. If the number of scan packets and the number of master bots are set higher than the threshold, duplicate infections can occur and malware can infect fewer devices than we expected. On the contrary, if the number of scan packets and the number of master bots are set below the threshold to reduce duplicate infections, the malware cannot be rapidly spread. In other words, it is necessary to set the number of master bots and scan packets to the extent that duplicate infections do not occur, considering the size of the infrastructure and the number of components. Therefore, malware attack experiments were performed on the internal network by changing the scan method, number of master bots, and number of scanning packets.

### 5.1. Experimental Environment

As shown in Figure 9 and Figure 10, real and virtual environments were constructed for the malware propagation experiment.

We built same environment as the test bed built to demonstration of the smart segmentation framework [13] according to the purpose of implementing the malware simulation tool. The actual environment has cost and space restrictions compared to the virtual environment, making it difficult to build large environments, but virtual environment is advantageous in terms of flexibility and scalability. In order to verify the functionality of the smart segmentation framework, a real environment same as the test bed to which the smart segmentation framework is applied was established, and a virtual environment was also established to enable experiments in a large network environment in the future.

In order to verify the functionality of the smart segmentation framework that quickly detect and block threats spreading to IoT infrastructure, the malware simulation tool had to spread malicious code as quickly and widely as possible. To do this, we had to find the optimal conditions for the malware simulation tool to spread malicious code as quickly and widely as possible to real environment. Therefore, experiments were conducted in real and virtual environments that did not adapt the smart segmentation framework in order to find the optimal conditions for the malware simulation tool to spread as quickly and widely as possible.

In a virtual environment, multiple virtual machines had to operate to configure the same environment as the real environment. Due to the limited specification of the host PC where the virtual machine operates, it was difficult to match the Emulated Server and Emulated IoT device specifications with the actual device used in the real environment. Therefore, it is important to note that the specifications of the two environments are not similar when looking at the experimental results.

#### 5.1.1. Real World

In the real world, the botnet infrastructure, such as the C&C server and DNS server, for malware attacks is connected to Gateway No. 1, while 45 Raspberry Pis are connected to Gateway No. 2, as shown in Figure 9. Raspberry Pi and Gateway communicate via Wi-Fi. That is, the internal network uses the IEEE 802.11 protocol. Moreover, the specifications of servers, gateways, and IoT devices used in real-world environments are as shown in Table 4.

#### 5.1.2. Virtual World

In the virtual environment, QEMU was used to emulate the C&C server, DNS server, and 45 Raspberry Pis within a single host PC. It established a virtual network environment and implemented similarly as in the real world, as shown in Figure 10. In addition, private IP addresses were assigned to emulated virtual devices, enabling them to communicate with each other. The specifications for each device used in the virtual-world environment are as shown in Table 5. It emulates C&C servers, DNS servers, and Raspberry Pi using QEMU emulator version 2.11.1. Furthermore, the 2018-06-27 Raspbian stretch lite OS image file and the 4.14.79-stretch QEMU kernel file were used to emulate IoT devices.

### 5.2. Performance Analysis

The appropriate number of scan packets should be known to avoid overloading network equipment when conducting malware propagation experiments in a local network environment. In addition, the appropriate number of master bots should be known to reduce duplicate infections with malware in IoT devices. Therefore, malware propagation experiments were conducted with one master bot to determine the impact of the number of scan packets on the total propagation time and the number of duplicate infections. Furthermore, malware propagation experiments were conducted with one scan packet to determine the impact of the number of master bots on the total propagation time and the number of duplicate infections.

#### 5.2.1. The Total Propagation Time: Malware Propagation Performance According to the Number of Scan Packets

Figure 11 shows the total propagation time and the number of duplicate infections per scanning method according to the number of scan packets. Comparing the total propagation time of the restricted random scanning method with the total propagation time of the sequential scanning method, the total propagation time of the sequential scan method was shorter. Furthermore, we can know that the total propagation time decreases as the number of scan packets increases. Note that if the number of scan packets exceeds 35, the total propagation time did not decrease even if the number of scan packets increased. This shows that in small–scale network environments, when performing malware attacks, the number of scan packets should be set to 30 or less.

We assumed that the divided scanning methods would reduce the total propagation time, but the experimental results did not. If the number of scan packets is less than five, the total propagation time is certainly short when using the divided scanning methods, but if the number of scan packets is more than five, the total propagation time of the divided scanning methods is not significantly different compared to the total propagation times of other scanning methods. In addition, comparing the total propagation time when the number of scan packets is less than 5 in the divided scanning methods and the total propagation time when the number of scan packets is more than 5 in the divided scanning methods, the total propagation time is short when the number of scan packets is more than 5. Therefore, using the restricted random scanning method and the sequential scanning method rather than the divided scanning methods is effective in reducing the total propagation time.

There are only two reasons why the graph shows experimental results with the number of scan packets from 1 to 45. First, when using the sequential and restricted random scanning methods, the total propagation time did not significantly decrease or increase even if the number of scan packets exceeded 45. In addition, when using the 6–division scan method, the number of duplicate infections surged when the number of scan packets exceeded 45, resulting in the problem of not infecting the 45 devices used in the experiment entirely. Because the total propagation time measures the time it takes for the entire 45 devices used in the experiment to be infected with malware, the total propagation time could not be measured if the number of scan packets exceeded 45 in the divided scanning methods.

#### 5.2.2. The Number of Duplicate Infections: Malware Propagation Performance According to the Number of Scan Packets

Figure 12 shows the number of duplicate infections per scanning methods according to the number of scan packets. The use of sequential or restricted random scanning methods is considered effective in reducing the number of duplicate infections rather than employing divided scanning methods when conducting malicious code diffusion experiments.

Furthermore, we can know that if the number of scan packets exceeds 30, the number of duplicate infections increases as the number of scan packets increases. In particular, when performing malware attacks in divided scanning methods, we can know a significant increase in the number of duplicate infections as the number of scan packets increased. When using the 6–divided scanning methods, these results are most noticeable because the 6–divided scanning methods scan for the smallest range of IP addresses among the proposed scanning methods.

Therefore, using the sequential or restricted random scanning methods and setting the number of scan packets below 30 is effective in reducing the number of duplicate infections.

In addition, we can know that there is a significant difference between experiments in the real environment for duplicate infections and those in the virtual environment. Comparing the two graphs in Figure 12, we can see that the number of duplicate infections increased significantly as the number of scan packets increased in the virtual environment. This seems to be due to the low hardware performance of the host PC running the virtual machine in the virtual environment. The performance of virtual networks in the host PC is not as supported as real network equipment, so the infection rate appears to be slower than the scan rate, resulting in duplicate infections.

The scan rate is the speed at which a master bot finds a vulnerable device, and the infection rate is the speed at which a vulnerable device runs malware and connects to the C&C server after the master bot injects malware into the vulnerable device. In the virtual environment, owing to the limited hardware performance of the host PC, it appears that vulnerable devices require more time to download malware from the HTTP server than in the real environment. Therefore, there appear to be many duplicate infections in the virtual environment, even if the number of scan packet is small because the infection rate is slower than the scan rate.

#### 5.2.3. Malware Propagation Performance According to the Number of Master Bots

Figure 13 shows the results of the total propagation time according to the number of master bots. The number of master bots reduces the total propagation time when using the restricted random scanning method and the divided random scanning methods. However, there is little change in the total propagation time when using the sequential scanning method and the divided sequential scanning methods. The biggest factor is that there is only one number of scan packets used in the experiment, so few duplicate infections have occurred.

In the malware propagation experiments according to the number of master bots, the number of scan packets is set to one to know the effect of master bots. Therefore, when using the random scanning method, even if the number of master bots increases, there is a low probability of scanning the same IP address at the same time. Furthermore, due to the nature of the sequential scanning method, which sequentially scans the range of IP addresses, it is difficult to occur duplicate infections. As a result, the total propagation time decreased as the number of master bots increased when using the random scanning method, and the total propagation time did not increase or decrease as the number of master bots increased when using the sequential scanning method.

#### 5.2.4. Comparison of the Total Propagation Time and the Number of Duplicate Infection

Figure 14 compares the total propagation time and the number of duplicate infections according to the number of scan packets. These results were obtained by conducting the malware propagation experiments employing all proposed scanning methods in the real environment and in the virtual environment. Figure 14 shows that duplicate infections affect the total propagation time. Therefore, choosing the number of scan packets with the least number of duplicate infections is the fastest way to spread malware.

## 6. Conclusions

### 6.1. Implications

The malware simulation tool proposed in this paper is implemented for the verification of security solutions that detect or prevent network threats to IoT infrastructure, especially the smart segmentation framework. The malware simulation tool was implemented based on Mirai. However, the Mirai’s propagation technique has five problems. Therefore, we suggested the restricted–range stateless scanning methods and the device–to–device command injection method to solve five problems. The restricted range stateless scanning method and the device–to–device infection method solve three problems: the propagation issue (Section 4.1.1) that cannot quickly infect IoT devices to specific infrastructure due to a wide range of IP addresses, the access issue (Section 4.1.2) that the malware loader cannot connect to IoT devices in an internal network, and the deployment issue (Section 4.1.5) that botnet infrastructure is required on each local network.

Then, we experimented by changing the number of scan packets and the number of master bots according to the four proposed scanning methods. We solved two problems through experiments to find the appropriate number of scan packets and master bots for the size of the infrastructure: the duplicate infection issue (Section 4.1.3) due to the number of scan packets that do not account for the range of IP addresses, and the overload problem (Section 4.1.4) that occurs due to all bots performing scanning operations. We believe that the malware simulation tool is able to verify the functionality of network–based security solutions that detect and block malware spreading to IoT infrastructure. However, there are not many techniques to exploit IoT devices, so we think malware attacks using the malware simulation tool will be easily detected.

Through experiments, we found that the most influential factor for total propagation time was the number of scan packets. Furthermore, we found that the total propagation time decreased as the number of scan packets increased. However, when the number of scan packets exceeded a certain number, the total propagation time did not decrease. Based on the results of the experiments, we judge that the appropriate number of scan packets, the number of master bots, and a scanning method can be selected considering the size of the infrastructure managed by the verification target. However, we do not know that the malware simulation tool is appropriate to verify a security solution that manages a large–scale infrastructure because we only conducted experiments on a small–scale network like a local network.

However, the malware simulation tool proposed in this paper succeeded in malware attacks on small–scale networks, such as local networks. In addition, we obtained experimental results that allow us to determine the number of scan packets, the number of master bots, and the scanning method to reduce duplicate infections and total propagation time. Therefore, we believe that the malware simulation tool can be used as the foundation for an integrated IoT malware simulation tool, which is one of the future works.

### 6.2. Future Works

Integrated IoT malware simulation tool. We will add several exploit techniques to the malware simulation tool in addition to the exploit method of Mirai. We will implement an integrated IoT malware simulation tool by analyzing scanning and attack techniques. Above all, we will spread malware in various ways by implementing attack vectors that exploit vulnerabilities of IoT devices. Currently, the malware propagation method is not flexible because it only attacks IoT devices that use vulnerable credentials of Telnet services. Therefore, we will make the several scenarios of spreading threats to infrastructure by implementing attack vectors. In addition to attack techniques of IoT devices, we will implement various characteristics of IoT malware. Furthermore, while only malware propagation is possible with the malware simulation tool so far, it can also be used as a DDoS attack tool through further research and functional implementation since it implemented based on Mirai malware. Integrated IoT malware simulation tool will allow experiments with a variety of malware attacks by combining the features of multiple IoT malware.

Automated and optimized simulation tool. We will implement an automated and optimized simulation tool. To date, we have implemented the four scanning methods on the malware simulation tool. However, setting these conditions is not yet automated, thus providing an inconvenience to modify and set conditions up individually when experimenting with other conditions. To solve this problem, we will implement a tool that automatically sets the optimal experimental conditions for the malware simulation tool. Therefore, we will implement an automated simulation tool that can set the scanning method, the number of scan packets, and the number of master bots through the GUI.

Network scan detection avoidance technique. We will implement the network scan detection evade technique in the malware simulation tool. Mirai performs port scanning to locate vulnerable devices, in which case they can be easily detected by security solutions that detect port scans. Therefore, we will implement the evade techniques of network scan in the malware simulation tool, so it will verify the security solutions that detect scanning behaviors.

Simulate on large networks and malware behavior dataset. We will carry out the malware simulation tool with multiple exploit methods in the simulation process to collect malware behavior datasets. Currently, malware propagation experiments have been carried out on small networks. Therefore, we will configure a large–scale network environment with the simulation process. And we will conduct malware propagation experiments on large–scale networks. It will be possible to compare the results of malware propagation in large–scale networks and the results of malware propagation in small–scale networks Furthermore, network traffic generated when conducting malware propagation experiments in large–scale network environments can be used as IoT malware behavior datasets. Operating the above–mentioned integrated IoT malware simulation tool in a simulation environment will allow more diverse IoT malware behavior datasets to be collected.

## Figures and Tables

**Figure 1 sensors-21-06983-f001:**
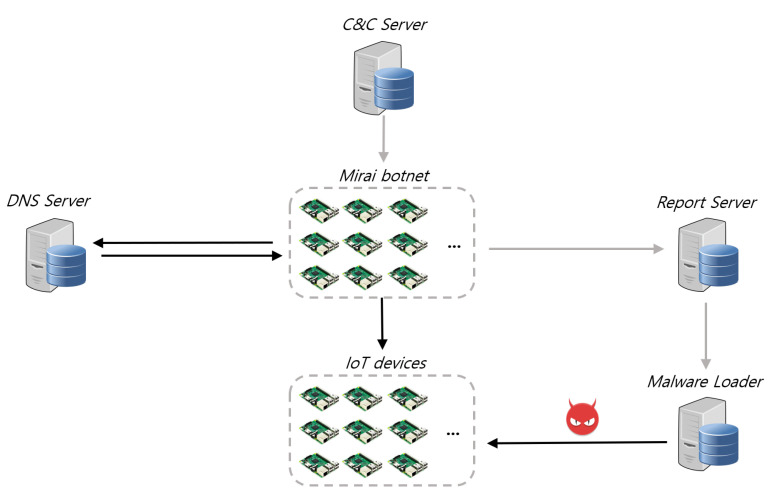
Mirai botnet configuration diagram.

**Figure 2 sensors-21-06983-f002:**
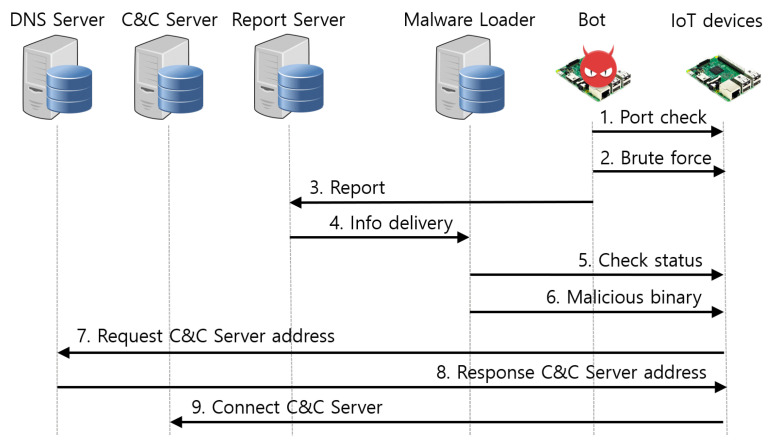
Mirai malware’s propagation process.

**Figure 3 sensors-21-06983-f003:**
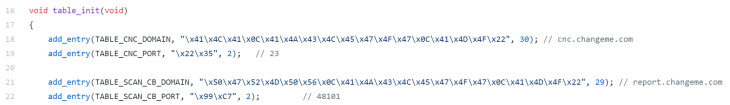
The domain names and port numbers of the C&C server and the report server (hardcoded to table_init function in table.c file among Mirai source code released on the Internet).

**Figure 4 sensors-21-06983-f004:**
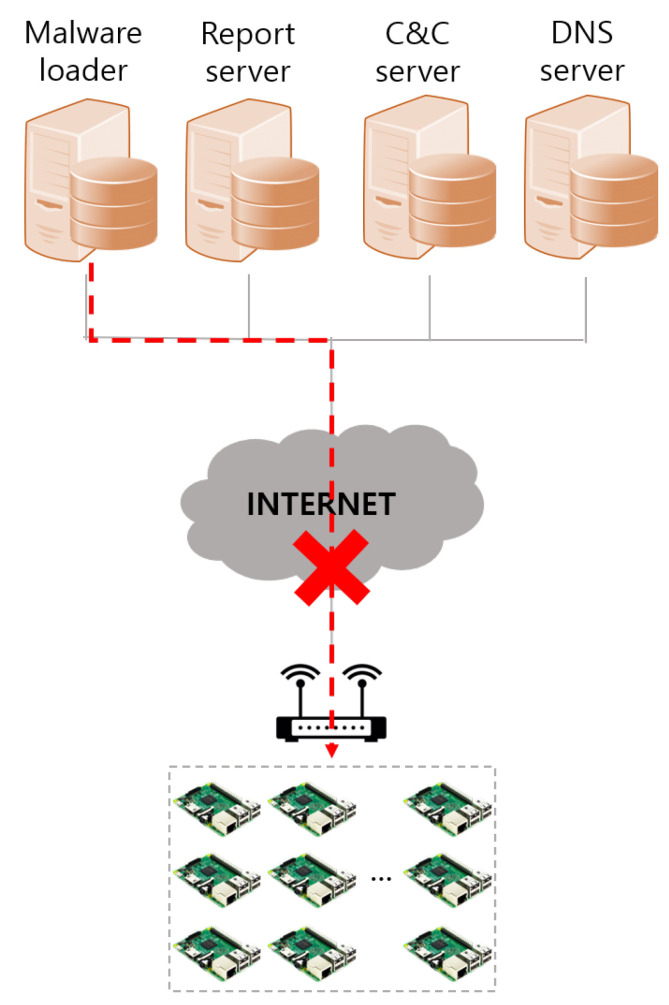
Inaccessible to internal network device.

**Figure 5 sensors-21-06983-f005:**
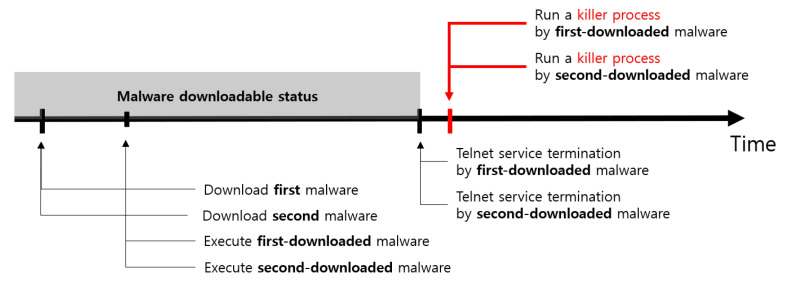
Timeline of malware download and execution.

**Figure 6 sensors-21-06983-f006:**
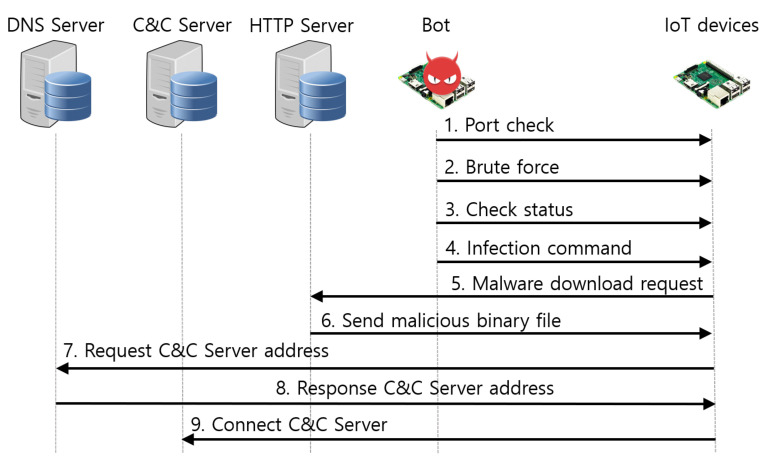
The stages of malware simulation tool.

**Figure 7 sensors-21-06983-f007:**
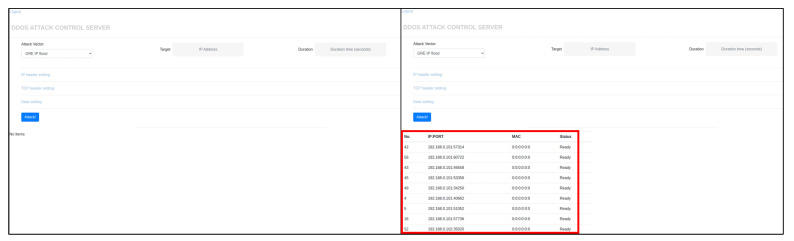
Before (**left**) and after (**right**) the spread of malware. Check the list of devices infected with malware on the CNC server administrator page.

**Figure 8 sensors-21-06983-f008:**
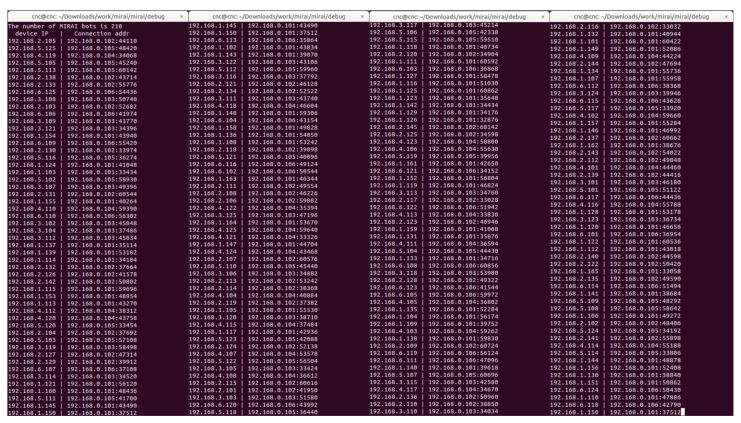
Check bot list information on the C&C server through Command Line Interface window.

**Figure 9 sensors-21-06983-f009:**
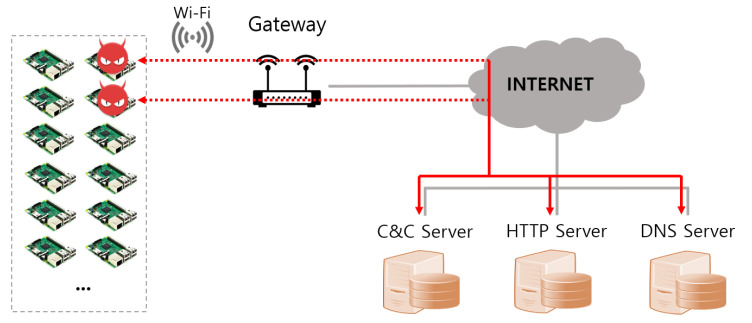
The real world for malware attack experiments.

**Figure 10 sensors-21-06983-f010:**
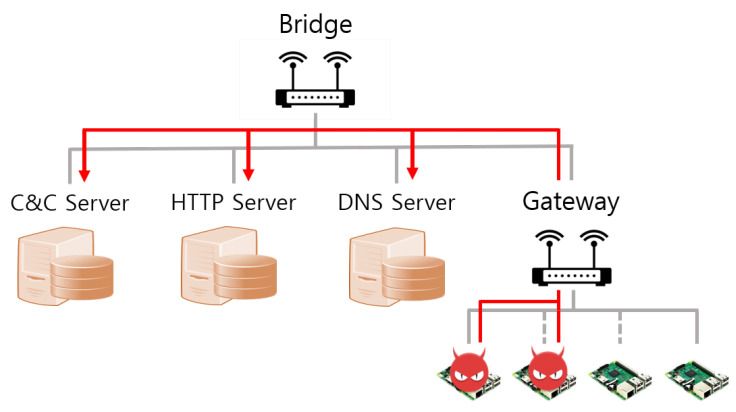
The virtual world for malware attack experiments.

**Figure 11 sensors-21-06983-f011:**
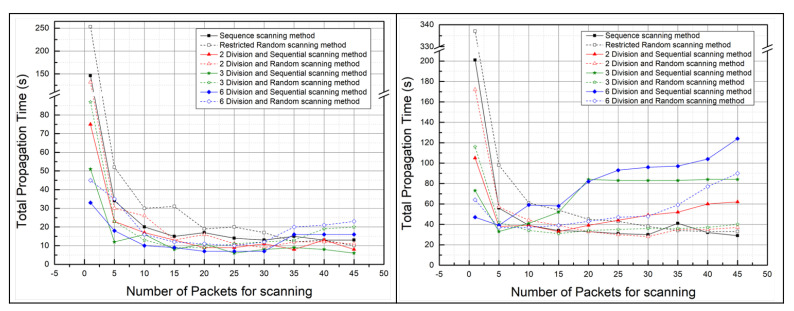
Total propagation time according to the number of scan packets in the real world (**left**) and in the virtual world (**right**).

**Figure 12 sensors-21-06983-f012:**
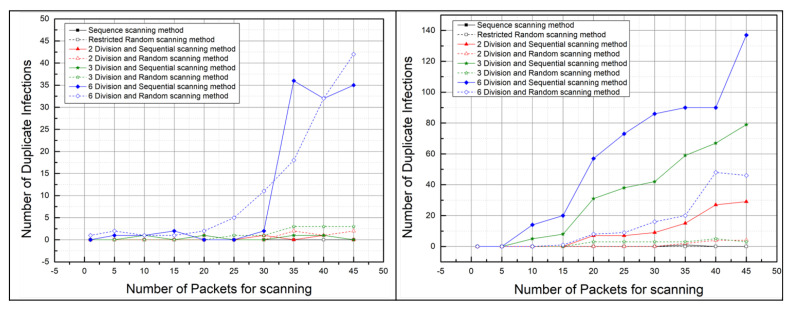
Number of duplicate infections according to the number of scan packets in the real world (**left**) and in the virtual world (**right**).

**Figure 13 sensors-21-06983-f013:**
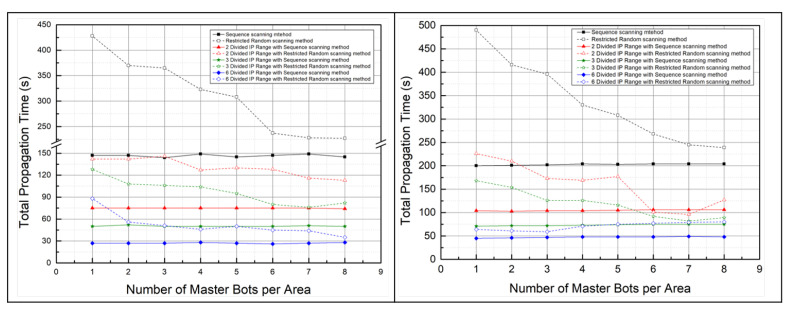
The total propagation time according to the number of master bots in the real world (**left**) and in the virtual world (**right**).

**Figure 14 sensors-21-06983-f014:**
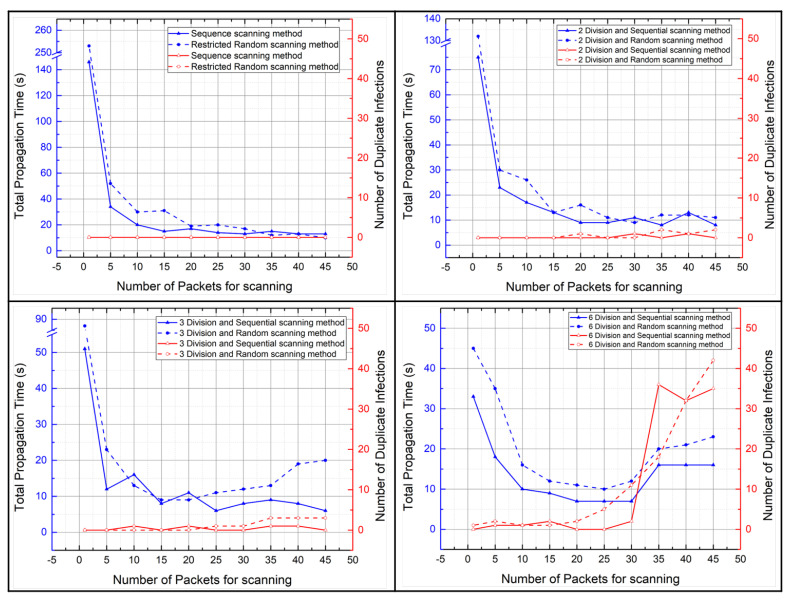
Total propagation time and the number of duplicate infections according to the number of scan packets in all proposed scanning method.

**Table 1 sensors-21-06983-t001:** A comparison between BASHLITE, Hajime, and Mirai.

Categories	BASHLITE (2014)	Hajime (2016)	Mirai (2016)
Target Arch.	MIPS, ARM	MIPS, ARM, x86/64	MIPS, ARM, x86/64
Capability	DDoS	not known	DDoS
Scan	Random	Random	Stateless, Random
Infection	Dictionary	Dictionary	Balanced
	password	password	dictionary password
Architecture	Centralized C&C	P2P	Centralized C&C
	(dedicated	(Decentralized)	(dedicated
	protocol)		protocol)
Persist	None	Anti reboot	Anti reboot
DGA	None	None	Use
Source code	[20]	not available	[10]

**Table 2 sensors-21-06983-t002:** A comparison between BrickerBot, IoT Reaper, and VPNFilter.

Categories	BrickerBot (2017)	IoT Reaper (2017)	VPNFilter (2018)
Target Arch.	BASH Script	Targeted	MIPS, ARM, x86/64,
		manufacturer	BASH Script
Capability	Destruction	DDoS	Destruction,
			Info. Theft,
			Man in the Middle,
			Resource abuse
Scan	Stateless, Random	Random	Stateless, Random
Infection	Balanced	Multiple CVE	Multiple CVE
	dictionary password		
Architecture	Centralized C&C	Centralized C&C	Centralized C&C
	(dedicated	(dedicated	(dedicated
	protocol)	protocol)	protocol)
Persist	None	Anti reboot	Anti reboot,
			Alter firmware
DGA	None	None	Use
Source code	not available	not available	not available

**Table 3 sensors-21-06983-t003:** Blacklist IP addresses.

Reason	IP Range
Loopback	127.0.0.0/8
Invalid address space	0.0.0.0/8
General Electric Company	3.0.0.0/8
Hewlett-Packard Company	15.0.0.0/7
US Postal Service	56.0.0.0/8
Internal network	10.0.0.0/8
	192.168.0.0/16
	172.16.0.0/14
IANA NAT reserved	169.254.0.0/16
	100.64.0.0/10
IANA Special use	198.18.0.0/15
Multicast	224.0.0.0-239.255.255.255

**Table 4 sensors-21-06983-t004:** Specification of devices used in the real-world environment.

	CPU	Intel(R) Core(TM) i7 CPU 960 @ 3.20 GHz
**Server**	RAM	12 G
	OS	Ubuntu bionic Ubuntu 18.04.5 LTS)
	CPU	ARMv7 Processor 5(v7l)
**Gateway**	RAM	497 MB
	OS	OpenWrt SNAPSHOT r13968-c5360894dc
	CPU	ARMv7 Processor 4 (v7l)
**IoT device**	RAM	1 GB
	OS	Raspbian GNU/Linux 9 stretch

**Table 5 sensors-21-06983-t005:** Specification of devices used in the virtual world environment.

**Host PC**	CPU	Intel(R) Core(TM) i9-10900 CPU
		@ 2.80 GHz, 20 Core
	RAM	128 GB
	OS	Ubuntu bionic Ubuntu 18.04.5 LTS)
**Emulated**	CPU	Intel(R) Core(TM) i9-10900 CPU
**Server**		@ 2.80 GHz, 4 Core
	RAM	8 GB
	OS	Ubuntu bionic Ubuntu 18.04.5 LTS)
**Emulated**	CPU	ARM1176
**IoT device**		(ARMv6-compatible processor rev 7 (v6l))
	RAM	253 MB
	OS	Raspbian GNU/Linux 9 stretch

## Data Availability

Data sharing not applicable.

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
