# Peer review of "A Malware Distribution Simulator for the Verification of Network Threat Prevention Tools"

_sensors, 2021, doi:10.3390/s21216983_

Round 1

Reviewer 1 Report

The topic presented is interesting, as well as the evaluation of propagation techniques. The main recommendation is to present the research from the perspective of the research process, where it is discussed in relation to the state of the art, hypothesis and methodology developed. For example, in the Related Works section the description of the botnets taken as a reference to design your proposal is presented, however, there is no possibility of comparing their results with a state of the art.

Authors are recommended to review the next format issues:Add corresponding capital letters in 257, 274, 296, 315 lines.

Replace the correct figure identifier in 371, 376, 390 and 398 lines.

I was wondering if it is possible to solve the following questions:

How is the risk of all components of the network calculated?

DPS has a trusted channel, what happens if it is attacked?

How do you calculate firmware and OS kernel integrity in the thing?

I would like to know what the characteristics of the devices and conditions are so that they are functional in a "Real World” the following processes:

  • Device-to-device(D2D)
  • Com mand injection (CI) propagation method
  • the bot takes over the role of the malware loader in the high-speed malware propagation tool
  • allows the bot to inject file download commands into vulnerable devices and execute injected commands by vulnerable devices  to request and download malware from the HTTP server
  • The vulnerable device uses the Wget command to request malware download to the HTTP server where the malware binary is stored

The main objective of the proposal was to scan internal networks, what happens with the protocols of internal networks?

In relation to the data link protocols, how does your proposal behave, do you have the same results for networks such as LoRaWAN, IEEE 802.11, etc?

It is not clearly observed what is the objective of testing the same experiments in a real environment and in a virtual one?

It is important to comment that the results of both environments are compared, and similar resources were not defined.

The tests consider small environments, what happens in larger networks, will you get similar results?

In real environments the infrastructure is greater, it is suggested to consider in the virtual environment or, failing that, to carry out a simulation process.

Reviewer 2 Report

This paper presents the results of some experiments on the spread of IoT malware  performed on simulated environment. The main contributions are listed on page 7. This in itself in a drawback of the paper. Place the contributions so late in the paper makes it harder for the reader to follow the tread of the paper. For the same reason, this paper would also benefit for an organising paragraph: (i.e. “the rest of this paper is organised as follows: Section 2 does… Sections 3 presents.., etc.)  Otherwise, its difficult for the reader to follow the paper.

For example, early, in the paper, it states that the  purpose of the propagation tool  (called a verification tool on line 54 )is to “to verify the functionality and performance of Smart segmentation solutions (SSS)”, and the reader expect to see results related to the effectiveness of one  or several malware detection tools. These are not present in the paper, only the propagation tool that can be used for such analysis.  Also, since SSS is never defined in the paper, a major benefit of the propagation tool is hazy.

It states in lines 62-62 that Mirai is the foundation of the malware propagation tool. But in the objectives on page 7  it states that  Mirai is not suitable to validate security tools, the reasons for which are listed in section 4. So its not clear in what sense Mirai is the foundation of the tool.

I’m not convinced problem 4.1.3 is a real problem. It is contingent on a very peculiar set of circumstances: a bot that can kill tetnet is installed, but before it does, a second bot is also installed and kills the first. The two would need to be very closely timed? How likely is that, when the paper just stated there are 3.6 B IP addresses?

Figure 5 not referenced in the text. Other figures not or barely explained. The concept of SSS is unclear, which makes one of the key contributions hard to grasp. Several figures and tables seems mislabeled.

The paper presents major lacunae in the level of English. Many sentences are unclear or ungrammatical, to the point where the meaning is unclear, and even the title seems ungrammatical. The paper would benefit from a complete revision by a native English speaker. In the edited pdf attached, I pointed out some of these unclear sentences, but many more require attention.

That said, the paper does make a valuable contribution to the field. The propagation tools is an inviable source of data for researchers, and can serve as the basis for much future research. I thus recommend that the paper be revised for English and organisation, and then resubmitted.

Reviewer 3 Report

This paper proposed a malware propagation tool that can spread the malicious software to internal network devices in an IoT infrastructure. Specifically, four scanning methods were implemented and results confirmed that increasing the number of packets for scanning reduced the total propagation time.  

Strength:
- The problem and motivation were well explained in the paper.
- The explored research is a very interesting and important topic

Weaknesses: 
- Methodology and experimental design should be improved. While the experiments were conducted to determine the appropriate number of packets to scan and it was concluded that the stability and speed of malware attacks are affected by the number of packets for scanning, it would be nice to see a detailed empirical analysis showing the impact of the packets that are sent over time and if the type of malware affects these results. Also, it would be nice to explore this capability on several types of IoT devices and prove that these results are consistent throughout. 
- Are there any other SSD security-based solutions? If so, how this approach is different from prior works? This is very important as it strengthens the contributions of the work.
- Related work focuses on BASHLITE, Hajime, Bricker, IoTReaper, and VPNFilter, but what about the Mirai malware? Does the type of malware can affect somehow the results? maybe as part of the future work, other malware and a variety of IoT devices can be used.

Round 2

Reviewer 2 Report

The authors have taken into account the comments I wrote in my initial review.

One issue I raised in my original review is that the tool described in this paper is confusingly  called both a “Propagation tool” and “verification tool” in different parts of the paper. Both appellations remain, causing confusion for the reader. The authors should information appellation. I also suggest that they consider the appellation: simulation tool.

I still believe that the paper would benefit form a review of the English language. The following are English language errors in parts of the text that has been added since the last version:

Line 68 : Introduce is the wrong word. Perhaps “explain” is what is meant.

Line 69 the malware à delete the

line 71 experimental environment à experimental environments

line 73 a malware propagation tool

Line 83: to comparing the differences between the Mirai malware and the malware

 propagation tool.--> Since you are comparing  a malware and malware propagation simulation tool, the differences are certainly night and day. You should qualify  this statement to say something like: comparing the differences between the Mirai malware and the malware

 propagation tool [with respect to scanning strategy …]

Line 87 have been assigned private IP addresses for spreading malware within IoT infrastructure à this is an awkward turn of phrase that seems to imply that the  private address was address assigned so that the device would in be infected.

Ine 88: the entire

89 device in the IoT infrastructure, à you probably mean « infect every device in the IoT infrastructire ».

Line 91 our implications : the implications of the research or the implications of our work etc.
